# A Temperature-Independent Methodology for Polymer Bitumen Modification Evaluation Based on DSR Measurement

**DOI:** 10.3390/polym14050848

**Published:** 2022-02-22

**Authors:** Jiantao Wu, Haoan Wang, Quan Liu, Yangming Gao, Shengjie Liu

**Affiliations:** 1College of Civil and Transportation Engineering, Hohai University, Nanjing 210098, China; jiantao.wu@hhu.edu.cn (J.W.); wanghaoan1998@outlook.com (H.W.); quan.liu@hhu.edu.cn (Q.L.); lsjwork@126.com (S.L.); 2Faculty of Civil Engineering and Geosciences, Delft University of Technology, 2628 CN Delft, The Netherlands

**Keywords:** bitumen modification, temperature-independency, complex modulus, dynamic shear rheometer, effective modification area, optimal modification index

## Abstract

Owing to the continuous increase of traffic loads, bitumen modification has been manifested as an efficient methodology to enhance asphaltic pavement performance. Currently, the modification index, defined as the ratio of mechanical properties (e.g., complex modulus) before and after bitumen modification, is extensively adopted to evaluate the modification degree. However, bituminous materials behave as temperature-dependent, which indicates that the mechanical property varies with measured temperatures. As a result, the calculated modification index also shows temperature-dependent property, which inhibits the use of modification index. For this reason, this study introduced a method to eliminate the temperature-dependency of the modification index. In specific, a mathematical model considering the properties of modifiers was firstly established to predict the modification index-temperature curve (MI-T curve). In what follows, the temperature-dependency of modification index was analyzed to verify the proposed model on three types of modifiers, which were graphene, Styrene-Butadiene-Styrene (SBS), and Ethyl-Vinyl-Acetate (EVA), respectively. The results indicated that the developed model could efficiently predict the MI-T curves. Besides, the effective modification area (EMA) and optimal modification index (OMI) were two reasonable indicators that evaluate the bitumen modification without considering the temperature-dependency.

## 1. Introduction

Bitumen modification is an increasingly important area as the demand for high-duty and long lifespan road construction is increasing [1,2]. For many years, researchers have been devoted to improving or modifying the bitumen performance. It requires a comprehensive demonstration to determine whether a particular substance is feasible for bitumen modification. By far, many studies have confirmed the success of various kinds of modifiers in the bitumen [3,4,5,6]. Currently reported modifiers with promising applications are polymers, chemical modifiers, recycled solid waste, and other additives (for instance, adhesion improvers and anti-oxidants) [7]. Nevertheless, it has not come to a consensus as to in which method the modification degree of bitumen can be well estimated. With the extensive use of the dynamic shear rheometer (DSR), Airey introduced the ratio of complex modulus before and after modification to evaluate the modification degree of bitumen [8]. However, this indicator did not take into account the temperature-dependency of bituminous binders. As a result, the indicator is calculated at a specific temperature. It is, therefore, not able to describe the modification degree in a broad temperature range. From the material aspect, the modified bitumen is a kind of composite material. Many theories, to date, have been developed to predict the complex behavior of a composite. The rule of mixtures (ROM) can predict various composite properties such as viscosity, elastic modulus, and tensile strength. Although the rule of mixtures is conventionally feasible for fibers, this conceptualization can be imitated and further extended to the modified bitumen. In fact, many existing methods for the property prediction of bituminous materials have already adopted the conceptualization. For example, the performance grade of new bitumen in reclaimed asphalt pavement was determined by a similar rule [9]. Besides, the prediction of bitumen viscosity also used a form of ROM [10]. To this end, this study evaluated the bitumen modification based on the DSR measurement. The modification index as a function of temperature (herein called the MI-T curve) can be captured by integrating the rule of mixtures. Following this, the experimental MI-T curves of different modified bitumen were used to verify the predicted MI-T curves. Consequently, two temperature-independent indicators were proposed to evaluate the modification degree of bitumen.

## 2. Development of MI-T Curves

### 2.1. The Rule of Mixtures (ROM)

The rule of mixture introduced a general rule for composite materials, which can predict various properties, such as elastic modulus and tensile strength [11]. The mathematical expression for *ROM* can be written as follows [12]:(1)$$E_{c}=fE_{f}+\left(1-f\right)E_{m}$$
where *f* is the volume fraction, *E*${}_{c}$, *E*${}_{f}$, and *E*${}_{m}$ represent the material properties of composite, fibers, and matrix, respectively. In modified bitumen, the bitumen can be regarded as the matrix, while the modifier was identical to the fibers, as shown in Figure 1. Although the modifier used might not be the fiber, previous studies have demonstrated ROM’s success in predicting bitumen’s viscosity [13]. Therefore, this study adopted the *ROM* to predict the complex modulus of modified bitumen.

### 2.2. MI-T Curve Based on ROM

The modification index based on DSR measurement can be expressed as Equation (Equation 2) [14]:(2)$$MI=G_{m}/G_{0}$$
where: *MI* is the modification index, *G*${}_{m}$ refers to the complex modulus of modified bitumen, and *G*${}_{0}$ refers to the complex modulus of the bitumen matrix.

When incorporating Equation (Equation 1) into Equation (Equation 2), the modification index can be expressed as Equation (Equation 4).
(3)$$G_{m}=G_{0}V_{0}+G_{f}\left(1-V_{0}\right)$$
(4)$$MI=V_{0}+\frac{G_{f}\left(1-V_{0}\right)}{G_{0}}$$
where: *V*${}_{0}$ refers to the volume fraction of the bitumen matrix, and *G*${}_{f}$ means the complex modulus of the modifier.

The complex modulus of bitumen is a function of the temperature, as shown in Figure 2. At low temperature (*T* < *T*${}_{g}$), bitumen exhibits a glassy elastic state (elastic modulus), and at high temperature (*T* > *T*${}_{g}$), it has a fluid dynamic (viscosity) state. From *T*${}_{g}$ to *T*${}_{f}$, the performance of bitumen approximates the viscoelastic state. *T*${}_{g}$ and *T*${}_{f}$ represent the two transformation temperatures in the phase transformation process of bitumen.

Herein, when the temperature is between *T*${}_{g}$ and *T*${}_{f}$, it is assumed that the complex modulus as a function of temperature approximates the linear relationship in the semi-log coordinate. In the conditions of temperature below *T*${}_{g}$ or above *T*${}_{f}$, the complex modulus is hypothesized constant. In this case, the complex modulus in a broad temperature can be expressed as Equation (Equation 5). As it can be seen from Equation (Equation 5), once the transformation temperature *T*${}_{g}$ and *T*${}_{f}$ are fixed, the complex modulus is a constant when the temperature is below *T*${}_{g}$ or above *T*${}_{f}$.
(5)$$G_{0}=\left\{\begin{array}{c} a_{0}10^{b_{0}T_{g}}\;T<T_{g}\\ a_{0}10^{b_{0}T}\;T_{g}\leq T\leq T_{f}\\ a_{0}10^{b_{0}T_{f}}\;T>T_{f} \end{array}\right.$$
where *a*${}_{0}$ refers to the constant related to the initial complex modulus at temperature 0 ${}^{\circ }$C, *b*${}_{0}$ indicates the temperature sensitivity of matrix bitumen within the temperature range as indicated in Figure 2. Concerning the complex modulus *G*${}_{f}$, numerous modifiers were reported in the bitumen modification. Some modifiers are distributed in bitumen in microparticles, such as mineral fillers (Figure 1a). In contrast, some presented in a network structure such as SBS (Figure 1b). Nevertheless, the morphology of modifiers in the bitumen will not be considered since this study aims to stress the feasibility of using ROM to illustrate the temperature-dependency phenomena. Alternatively, whether the modifier’s stiffness would alter with the temperature concerns the modification index’s calculation. Accordingly, two categories of modifiers were considered in this study. The first category are inert modifiers whose stiffness would not change with temperature. The other category represents the active modifier, whose stiffness is temperature-dependent. Similar to the complex modulus of the bitumen matrix, the complex modulus, therefore, can be written as follows.
(6)$$\begin{array}{c} \;\mathrm{Inert}:\;\;G_{f}==G_{f0}=C\\ \;\mathrm{Active}:\;\;G_{f}=\left\{\begin{array}{cc} a_{f}10^{b}f^{T}, & T<T_{gf}\\ a_{f}10^{b_{f}T_{gf},} & T\geq T_{gf} \end{array}\right. \end{array}$$
where *a*${}_{f}$ refers to the constant related to the modifier’s initial complex modulus at temperature 0 ${}^{\circ }$C, *b*${}_{f}$ refers to the temperature sensitivity of the modifier that was similarly defined with the sensitivity of bitumen, and *T*${}_{gf}$ represents the transition temperature of modifiers. By combining Equations (4)–(6), the modification index (*T* > *T*${}_{g}$) can be deduced, as seen in Table 1.

## 3. Properties of MI-T Curves

According to Table 1, theoretical MI-T curves can be plotted considering different parameter combinations. The initial parameters of bitumen were derived from experimental data. Below are three cases discussed in this section concerning *C*${}_{1}$ to *C*${}_{3}$ in Table 1.

Case 1: *V*${}_{0}$ = 0.8, *a*${}_{0}$ = 2 × 10${}^{7}$.

Based on Equation (Equation 7), theoretical MI-T curves are shown in Figure 3 and Figure 4. In general, for modified bitumen with inert modifiers, the modification index increased with the temperature. The transition temperature of bitumen *T*${}_{f}$ determined the curve shape. When the *T*${}_{f}$ was within the investigated temperature range, a platform would arise. Beyond the *T*${}_{f}$, the complex modulus of bitumen and modifier both approximated to a constant and, therefore, the modification index remained unchanged. Another crucial parameter is the *b*${}_{0}$, which represents the temperature sensitivity of bitumen. As the temperature sensitivity of bitumen increased, the modification index showed a more significant increasing rate with the temperature.

Case 2: *V*${}_{0}$ = 0.8, *a*${}_{0}$ = 4 × 10${}^{7}$, *T*${}_{f}$ = 30, *T*${}_{gf}$ = 80.

The MI-T curves for Case 2 are presented in Figure 5. It is worth noting that investigated temperature was limited to 70 ${}^{\circ }$C, although the *T*${}_{gf}$ reached 80 ${}^{\circ }$C. When the temperature exceeded the *T*${}_{gf}$, the formula in Equation (Equation 8) was identical to the one in Equation (Equation 9). Therefore, the temperature was constrained for the benefit of illustration.

In this case, the initial complex modulus of bitumen was 4 × 10${}^{7}$. Meanwhile, the initial complex modulus of the modifier considered two conditions. If the complex modulus of the modifier was lower than that of bitumen, the modifier was called the soft modifier. In contrast, the one with a higher complex modulus than bitumen was the hard modifier. Therefore, the initial complex modulus of modifier was fixed at 2 × 10${}^{7}$ to represent the soft modifier. The initial complex modulus at 6 × 10${}^{7}$ indicated that the modifier used had a higher stiffness than bitumen. Besides, the temperature sensitivity of the bitumen and modifier were taken into account to draw the MI-T curves.

It can be observed that the modification index of modified bitumen with a soft modifier was below 1.0. It complied with the practical experience that adding soft matter in the bitumen would reduce the complex modulus of bitumen. On the other hand, the modification index of bitumen modified with a hard modifier was higher than 1.0 when the temperature was relatively low. However, as the temperature increased, the modification index could decrease below 1.0. This is ascribed to the distinct temperature sensitivities for bitumen and modifier. Although the modifier was stiffer than bitumen at low temperature, faster attenuation of the complex modulus resulted in the modification index being lower than 1.0 at a specific temperature. For this reason, in the practice of bitumen modification, thermal stability is a critical factor when selecting the modifier.

The distinct temperature sensitivity essentially altered the MI-T curve shape. It can be intuitively seen that the modification index was no longer monotonous. When $b_{f}$ < $b_{0}$, the modification index increased to the maximum value and subsequently showed a decreasing tendency. When *b*${}_{f}$ > *b*${}_{0}$, the modification index decreased to a specific temperature, after which the decreasing rate was accelerated. It can be concluded that the transition temperature of bitumen determined where the maximum value was located.

Case 3: *V*${}_{0}$ = 0.8, *a*${}_{0}$ = 4 × 10${}^{7}$, *T*${}_{f}$ = 80, *T*${}_{gf}$ = 30.

The third case discussed in this study was calculated based on Equation (Equation 9), as shown in Figure 6. Similarly, the modifier’s stiffness was divided into soft and hard substances, represented by the initial complex modulus of 2 × 10${}^{7}$ and 6 × 10${}^{7}$, respectively. Unlike Case 2, the soft modifier could also show a modification index higher than 1.0, as seen in the continuous increase with the temperature when the *b*${}_{f}$ < *b*${}_{0}$. The comparison between *b*${}_{f}$ and *b*${}_{0}$ could predict the tendency of the modification index as the temperature increased.

## 4. Experimental Verification of MI-T Curves

Aiming to verify the proposed model in Table 1, experimental data were collected from previous studies. Three kinds of modified bitumen, including graphene-modified bitumen, SBS modified bitumen, and EVA modified bitumen were used to verify the model. Among which, SBS modified bitumen was measured by authors. The graphene-modified bitumen and EVA modified bitumen data were sourced from Yang et al. [14] and Airey [8], respectively. The MI-T curves of different modified bitumen are presented in Figure 7. As indicated, for each kind of modified bitumen, there are two dosages considered. The graphene-modified bitumen considered the amount of 0.8% and 1.0%, while EVA’s content was 5% and 7%, respectively. As for the SBS modified bitumen, the incorporation mass was 5% and 7%. It can be found that the modifier content would not essentially change the MI-T curve but move the MI-T curve upward or downward. Therefore, the effect of volume fraction on the shape of the MI-T curve was not discussed in detail.

The EVA modified bitumen showed single-peak MI-T curves, which corresponded to Case 2. Accordingly, the transition temperature *T*${}_{f}$ of bitumen was around 55 ${}^{\circ }$C, which was quite close to the softening point of bitumen. However, further investigation on the relationship between the transition temperature and the softening point of bitumen is required before achieving this conclusion. There are two places of experimental MI-T curves deviating from the predicted curves in terms of the graphene-modified bitumen. It also complied with the single-peak curve. However, in the increasing phase (10 ${}^{\circ }$C to 60 ${}^{\circ }$C), the MI-T curve approximated a linear relationship while, in predicted ones, the increasing rate was increased as the temperature increased. It might be attributed to the drawback of the proposed model since the complex modulus as a function of temperature was simplified. The other abnormal place is the presence of a single peak. According to the study of Yang et al. [14], the graphene used belongs to the carbon nanomaterials in the form of particles. The complex modulus should not considerably change with the temperature. In this sense, the graphene-modified bitumen should be classified into Case 1 rather than Case 2. The plausible explanation was on account of the excellent thermal conductivity of graphene. As a result, the complex modulus of bitumen as a function of temperature was significantly changed. The analysis above demonstrated that the proposed model in Table 1, to some extent, can be used to describe the MI-T curve of modified bitumen. At least, the model successfully predicted typical MI-T curves considering the properties of the modifier. Nevertheless, some deviations between the experiment and calculation were also observed. These deviations were sourced from the simplification in the model derivation process and inadequate consideration of bitumen–modifier interaction.

The modification index of SBS modified bitumen increased as the temperature increased when the temperature was below 70 ${}^{\circ }$C, following which, the modification index tended to be constant. Therefore, the SBS bitumen can be classified into Case 1 in Section 3. It is consistent with the condition of Case 1. SBS is a kind of rubber elastomer, showing low sensitivity of stiffness to the temperature change. Within the investigated temperature from 10 ${}^{\circ }$C to 90 ${}^{\circ }$C, the stiffness, therefore, can be regarded as unchanged.

## 5. Proposed Indicators Capable of Describing Overall Modification Degree

The aforementioned MI-T curves illustrated that the modification index is a highly temperature-dependent indicator. More importantly, there probably exists a maximum value of modification index in the investigated temperature range. In this case, a single modification index at a specific temperature was no longer representative for the modification degree of modified bitumen. For this reason, this study proposed using two indicators that are independent with temperature to feature the overall modification degree for modified bitumen, as shown in Figure 8. As illustrated in Figure 8, the effective modification area (EMA) and optimal modification index (OMI) can be mathematically expressed by Equations (10) and (11) as follows:(10)$$\mathrm{EMA}=\int_{T_{1}}^{T_{2}}\left(MI-1\right)dT$$
(11)$$\mathrm{OMI}=MI|T=T_{f}$$
where $T_{1}$ and $T_{2}$ represent the lower and upper limits of the temperature range, and *T* is the temperature. *T*${}_{f}$ has the same meaning as previously defined.

According to Figure 8, EMA can evaluate the modification degree considering a specific temperature range. In this case, the overall modification effect can be featured rather than at a specific temperature. A large EMA value suggests a better modification effect, vice versa. As for the OMI, it is only applicable when the MI-T curve shows a single peak form, which indicates the optimal modification effect where the modifier plays its best role in the bitumen. According to the analysis above, the single peak shows when the modifier is active and the transformation temperature of the modifier is higher than that of bitumen.

Table 2 shows the calculated EMA and OMI for six investigated modified bitumen with a temperature range from 10 ${}^{\circ }$C to 90 ${}^{\circ }$C. EMA can effectively identify the influence of mass fraction on the modification degree. The advantage of using EMA as the modification degree is that the modification degree can be featured over a temperature range. As for the OMI, it is only applicable for Case 2 introduced above.

## 6. Conclusions and Outlook

Based on the rule of mixture, the MI-T curves of modified bitumen were deduced. Two temperature-independent indicators, namely EMA and OMI, were proposed to describe the modification degree. It has been demonstrated that the proposed model can efficiently predict the MI-T curves of modified bitumen. The experimental MI-T curves can identify the modifier’s properties, thus better understanding the modification mechanism.

Nevertheless, the proposed model can only qualitatively describe the curve shape. To achieve the quantitative prediction, the interaction between bitumen and the modifier should be taken into account. Future effort should be made in model regression to obtain the model parameters. Besides, the morphology of modifiers in bitumen should also be considered to modify the MI-T curve. With the single peak MI-T curve, it would be promising to study the transition temperature of bitumen, which is most likely related to the softening point.

## Figures and Tables

**Figure 1 polymers-14-00848-f001:**
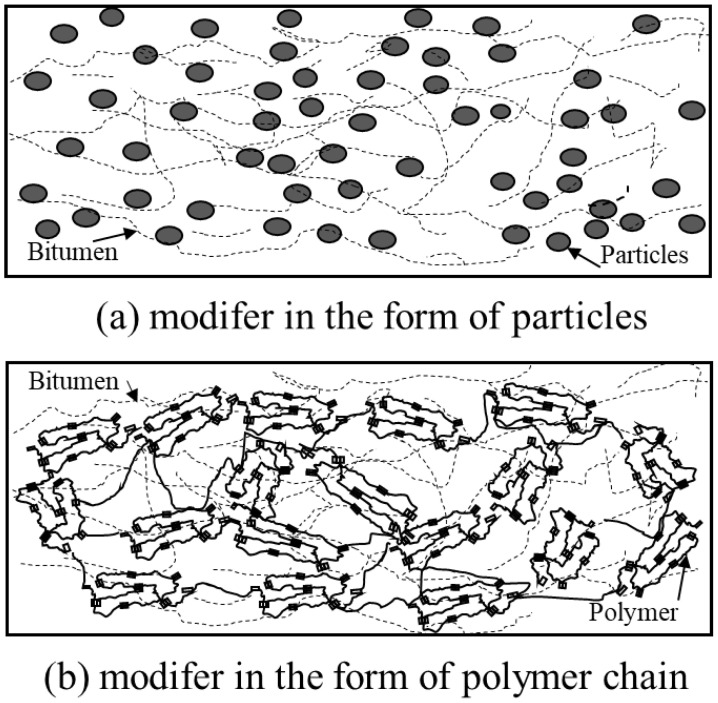
Schematic of the modified bitumen structures. (**a**) Particle, (**b**) polymer chain.

**Figure 2 polymers-14-00848-f002:**
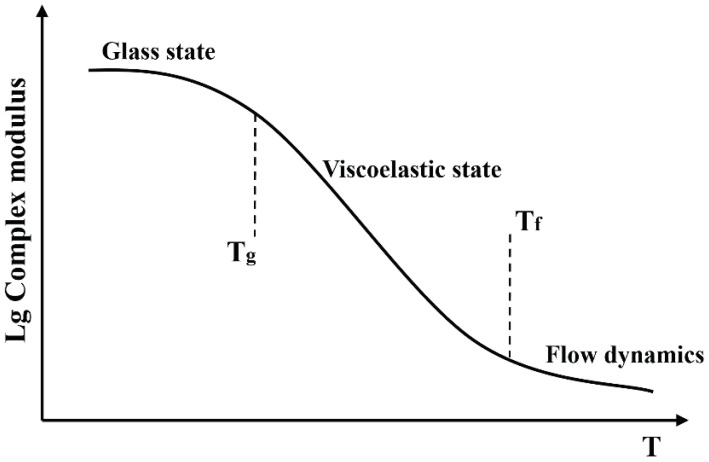
Complex modulus of bitumen as a function of temperature.

**Figure 3 polymers-14-00848-f003:**
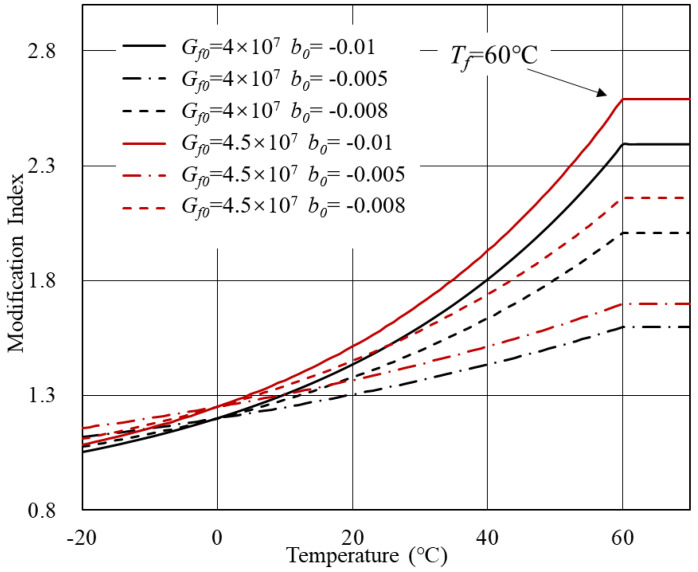
MI-T curves for Case 1 at *T*${}_{f}$ = 60 ${}^{\circ }$C.

**Figure 4 polymers-14-00848-f004:**
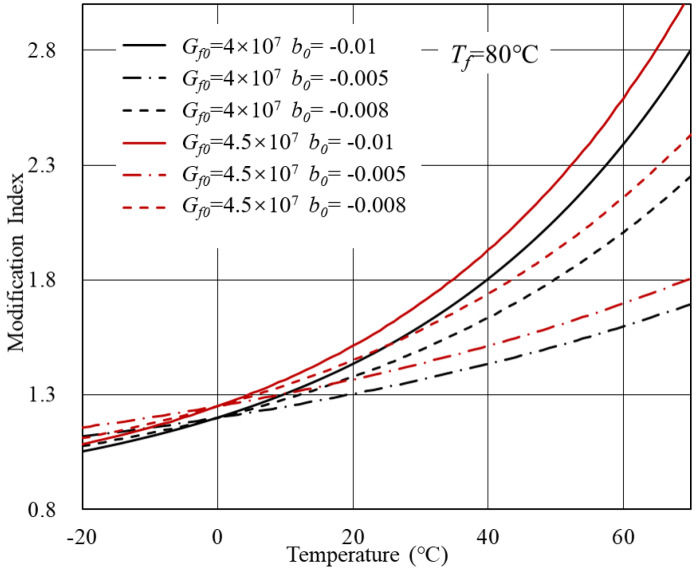
MI-T curves for Case 1 at *T*${}_{f}$ = 80 ${}^{\circ }$C.

**Figure 5 polymers-14-00848-f005:**
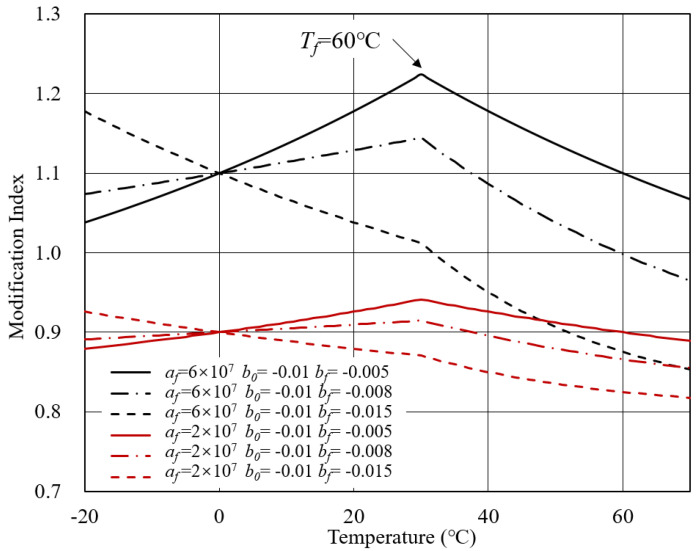
MI-T curves for Case 2.

**Figure 6 polymers-14-00848-f006:**
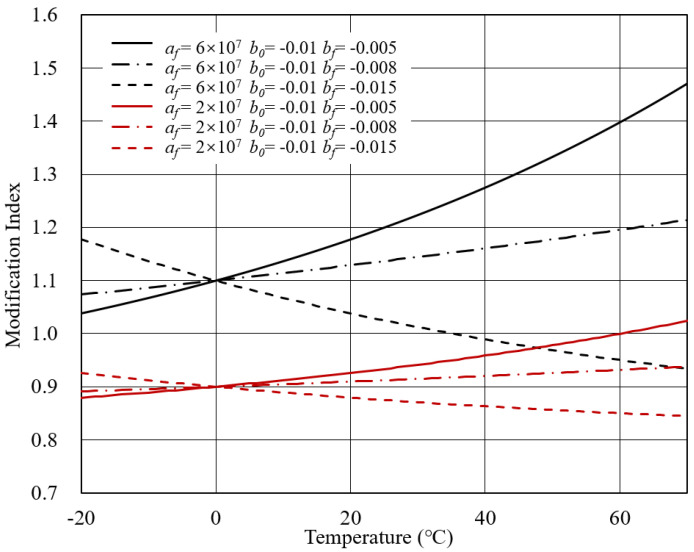
MI-T curves for Case 3.

**Figure 7 polymers-14-00848-f007:**
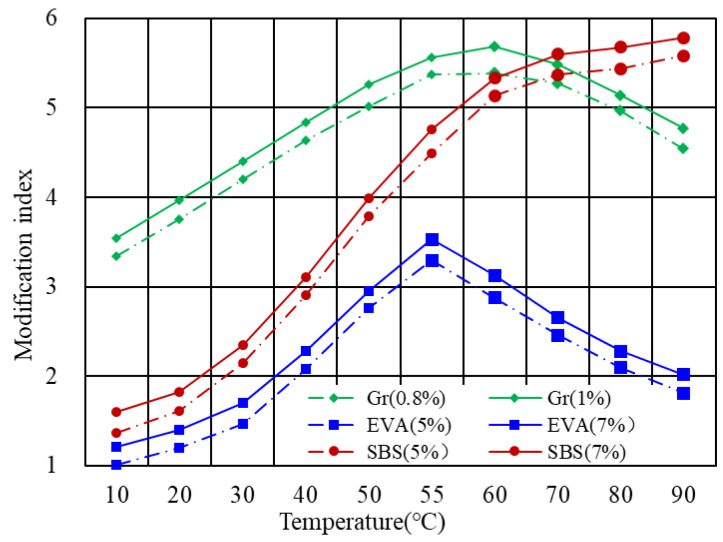
Experimental MI-T curves from literature.

**Figure 8 polymers-14-00848-f008:**
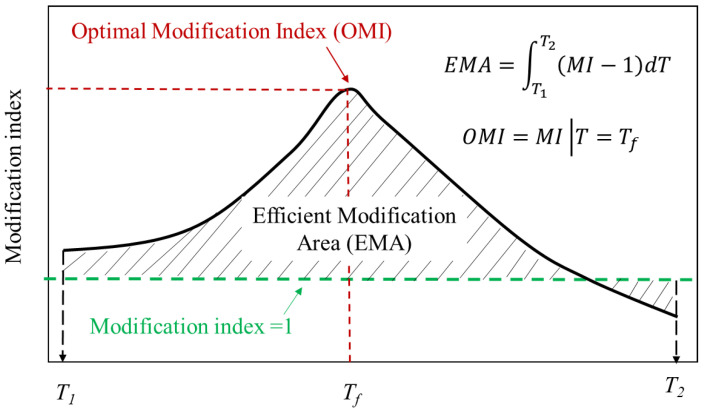
Schematic of EMA and OMI.

**Table 1 polymers-14-00848-t001:** The expression of the modification index.

Cases	Expressions
C1: inert modifier	(7) $$MI=\left\{\begin{array}{c} V_{0}+\frac{G_{f0}\left(1-V_{0}\right)}{a_{0}10^{b_{0}T}},T\leq T_{f}\\ V_{0}+\frac{G_{f0}\left(1-V_{0}\right)}{a_{0}10^{b_{0}T_{f}}},T>T_{f} \end{array}\right.$$
C2: active modifier, $T_{f}$ < $T_{g}f$	(8) $$MI=\left\{\begin{array}{c} V_{0}+\frac{a_{f}10^{b_{f}T}\left(1-V_{0}\right)}{a_{0}10^{b_{0}T}},T\leq T_{f}\\ V_{0}+\frac{a_{f}10^{b_{f}T}\left(1-V_{0}\right)}{a_{0}10^{b_{0}T_{f}}},T_{f}\leq T\leq T_{gf}\\ V_{0}+\frac{a_{f}10^{b_{f}T_{gf}\left(1-V_{0}\right)}}{a_{0}10^{b_{0}T_{f}}},T>T_{gf} \end{array}\right.$$
C3: active modifier, $T_{f}$ > $T_{g}f$	(9) $$MI=\left\{\begin{array}{c} V_{0}+\frac{a_{f}10^{b_{f}T}\left(1-V_{0}\right)}{a_{0}10^{b_{0}T}},T\leq T_{gf}\\ V_{0}+\frac{a_{f}10^{b_{f}T_{gf}\left(1-V_{0}\right)}}{a_{0}10^{b_{0}T}},T_{gf}\leq T\leq T\\ V_{0}+\frac{a_{f}10^{b_{f}T_{gf}\left(1-V_{0}\right)}}{a_{0}10^{b_{0}T_{f}}},T>T_{f} \end{array}\right.$$

**Table 2 polymers-14-00848-t002:** Calculated EMA and OMI for modified bitumen.

Type of Bitumen	EMA	OMI
Gr (0.8%)	292.46	–
Gr (1.0%)	309.68	–
EVA (5%)	85.88	–
EVA (7%)	102.52	4.50
SBS (5%)	218.92	5.38
SBS (7%)	236.15	–

## Data Availability

Data is contained within the article.

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
