# Peer review of "A Temperature-Independent Methodology for Polymer Bitumen Modification Evaluation Based on DSR Measurement"

_polymers, 2022, doi:10.3390/polym14050848_

Round 1

Reviewer 1 Report

General comments

Structure of the article (Identification of the gap in knowledge): In this study, the gap in knowledge is the consideration of temperature dependency of modified bituminous binders in order to assess the modification degree of bitumen.

References: Most of the cited references are not updated (before 2017-18). . The article does not have an abnormal number of self-citations.

Relation hypothesis vs experimental design: The hypothesis was not clearly defined in the article so it was not possible to compare the experimental design.

Reproducibility: The article supplies sufficient information to reproduce the results.

Figures/Tables/Images/Schemes: They properly show the data, and they are easy to interpret and understand.

Conclusions: Conclusions are consistent and coherent with the analysis and results obtained.

Specific comments

- Pages 1 and 2 Lines 37, 38

Please add more information about the extension of the use of ROM for modified Bitumen. Since bitumen is a very complex material it is important to include more references and explanations to check if the generalization can be feasible.

- Page 3 Equation (3)

Gf means the complex modulus of the modifier. Please include more information about this. Adding more references about this parameter would be valuable.

- Page 3 Lane 68 – 69 Equation (5)

Complex Modulus is hypothesized constant, but equation (5) shows that below Tg and above Tf, the complex modulus is affected by temperature. Please add more information. or explain this.

- Page 3 Line 77

Please explain why the morphology of the modifiers is not a concern in your study. It could be relevant for the mixture.

- Page 4 Line 85

Bo, Bf refers to the temperature sensitivity of the modifier. Please add information about this parameter.

- Page 5 Figure 3 and 4

How do you define Tf=60°C and Tf=80°C

- Pages 8 and 9

Please include more information about the proposed index EMA and OMI. It is important to show the criteria to evaluate these indices. Why OMI only applies to EVA (7%) and SB (5%)?

Reviewer 2 Report

Dear authors,

interesting paper about predictions and evaluations done in respect to better identify modification index. The approach to extend a single point index by a curve showing also temperature dependency which is for bitumen and several modifiers quite important is useful. In general I have no comments or questions to the overall concept you proposed and described. It was clearly presented. There are few minor issues to be addressed.

Line 27: within the modifiers you are mentioning adhesion promoters as well. I have long-termly a bit reserved opinion about how to classify adhesion promoters. For me they are not modifiers. Bitumen is usually just carrying the compound which acts on the interface bitumen-aggregate to allow better and more durable coating of the aggregate particles by bitumen. My opinion is that this is not a true bitumen modification. What is your opinion?

Line 80: the first "is" shall be in my opinion "are".

Presenting the cases C1 to C3, shall the values of complex modules not be presented with some units? And the values you are considering in the cases like 2e7 etc., they are related to some particular bitumen (paving grade)? Can this probably be better specified? I mean this value can be OK and true e.g. for 50/70, but if you would use 70/100 or a different grade it might change. Was the research done and validated by using one paving grade or several?

In the text terms "soft" and "hard" modifiers are used. What is the parameter or set of parameters which differentiate what is a oft or a hard modifier? Can this be maybe a bit as a short explanation included in the text?

Line 170: change "pro-posed" to "proposed". 
